# Volumetric Differences in Cerebellum and Brainstem in Patients with Migraine: A UK Biobank Study

**DOI:** 10.3390/biomedicines11092528

**Published:** 2023-09-13

**Authors:** Oreste Affatato, Gull Rukh, Helgi Birgir Schiöth, Jessica Mwinyi

**Affiliations:** 1Functional Pharmacology and Neuroscience Unit, Department of Surgical Science, Uppsala University, 752 36 Uppsala, Sweden; 2Uppsala University’s Centre for Women’s Mental Health during the Reproductive Lifespan—WoMHeR, Uppsala University, 752 36 Uppsala, Sweden

**Keywords:** migraine, cerebellum, brainstem, structural MRI, UK Biobank

## Abstract

**Background:** The cerebellum and the brainstem are two brain structures involved in pain processing and modulation that have also been associated with migraine pathophysiology. The aim of this study was to investigate possible associations between the morphology of the cerebellum and brainstem and migraine, focusing on gray matter differences in these brain areas. **Methods:** The analyses were based on data from 712 individuals with migraine and 45,681 healthy controls from the UK Biobank study. Generalized linear models were used to estimate the mean gray matter volumetric differences in the brainstem and the cerebellum. The models were adjusted for important biological covariates such as BMI, age, sex, total brain volume, diastolic blood pressure, alcohol intake frequency, current tobacco smoking, assessment center, material deprivation, ethnic background, and a wide variety of health conditions. Secondary analyses investigated volumetric correlation between cerebellar sub-regions. **Results:** We found larger gray matter volumes in the cerebellar sub-regions V (mean difference: 72 mm${}^{3}$, 95% CI [13, 132]), crus I (mean difference: 259 mm${}^{3}$, 95% CI [9, 510]), VIIIa (mean difference: 120 mm${}^{3}$, 95% CI [0.9, 238]), and X (mean difference: 14 mm${}^{3}$, 95% CI [1, 27]). **Conclusions:** Individuals with migraine show larger gray matter volumes in several cerebellar sub-regions than controls. These findings support the hypothesis that the cerebellum plays a role in the pathophysiology of migraine.

## 1. Introduction

The cerebellum is a highly organized and major structure of the hindbrain, tightly connected to the cerebrum, brainstem, and spinal cord [1]. The cerebellum is a fundamental center for motion control and coordination [2], involved also in processing information at affective and cognitive levels [3]. Recent studies demonstrated the involvement of the cerebellum in nociception and pain modulation, highlighting a possible connection with migraine [4,5]. Moreover, the cerebellum is innervated by the trigeminovascular complex, a structure that plays a major role in migraine pathophysiology [1]. This complex also projects in the brainstem, another important brain structure involved in pain perception and modulation as well as migraine pathogenesis [6]. Thus, the cerebellum and the brainstem are two major brain regions that play a considerable role in migraine onset and progression, although the exact pathophysiological mechanism is still under debate. Importantly, large-scale studies investigating to what extent the involvement of the brainstem and cerebellum in migraine pathogenesis may be reflected in brain morphological changes are missing.

Structural magnetic resonance imaging (MRI) is a powerful tool used to assess possible associations between volumetric differences in particular brain regions and the clinical presentation of brain-related disorders. Structural MRI can reveal volumetric alterations in certain areas of the brain that may reflect local dysfunction or atrophy, thus helping to elucidate the possible pathological pathway of the disorder under investigation. Hitherto, only a few studies of overall small sample sizes have studied volumetric differences in the brainstem or cerebellum between individuals with and without migraine. One of the earliest migraine-related MRI studies assessed structural and functional divergences in brain volumes between patients with migraine without aura and healthy controls [7]. The study comprised 42 participants in total, equally distributed between cases and controls. The authors found that individuals with migraine exhibited lower cerebellar and brainstem volumes than controls. Another study used structural MRI techniques to assess volumetric changes between individuals with chronic migraine and healthy controls [6]. The study comprised 24 women with chronic migraine and 24 controls, matched by age and sex. Also, this study observed lower brainstem and cerebellar volumes. A subsequent study addressed gray matter volume differences between chronic migraine individuals and controls [8]. They performed the brain scan on a total of 40 participants, equally divided into migraine cases and healthy controls, and found an overall lower gray matter volume in individuals with migraine. In particular, they found a volume reduction in the right cerebellum, in the crus II and in the lobule VIIIa, an area known to be involved in trigeminal nociception [9]. These findings were also supported by another study that used tailored analytical techniques to assess brain volumetric differences [10]. However, another study detected increased gray matter volumes in the cerebellum at the level of crus I and II and the regions VI, VIIb, VIIIa [11], partly contradicting the results obtained in previous studies [9,10]. Overall, these studies highlighted some major patterns at the level of brain morphology and provided some evidence for the possible involvement of cerebellar and brainstem areas in migraine pathophysiology. In particular, the lower gray matter volumes in certain regions could be due to local dysfunction or atrophy that might in turn lead to increased migraine or headache susceptibility.

The mentioned studies, however, are generally characterized by small sample sizes. It is also noteworthy that the statistical analyses adopted in the majority of these studies rather relied on tests such as *t*-tests or non-parametric versions of the same test family, not correcting for confounding. Moreover, the t statistic is a dimensionless measure of a mean difference between two populations and therefore it is difficult to interpret it in a biologically meaningful way, especially in relation to MRI measurements. More sophisticated methods, such as generalized linear regression models, can robustly estimate the mean difference in volumes of a target brain region, expressed in more concrete units of measure. This can grant a better evaluation of the biological implication of such volumetric differences. Moreover, the generalized linear regression allows for the inclusion of a higher variety of predictors in the model, ensuring a less biased estimation of the population parameters.

The aim of this study is therefore to estimate brain volumetric differences in the cerebellum and the brainstem between patients with migraine and healthy controls, using generalized linear models and data from the large UK Biobank cohort. Including more than 700 individuals with migraine and more than 45,000 controls, this comprehensive dataset can ensure a higher precision due to the large sample size. Moreover, conditioning our regression models with a complete set of demographic and biological variables allows for a less biased estimation of the mean volumetric differences between cases and controls.

## 2. Materials and Methods

We used data collected from the UK Biobank, a large population-based cohort with approximately half a million participants from all across the United Kingdom. The UK Biobank repository gathered in-depth information from all the volunteers, from lifestyle phenotypes to genetic and neuroimaging data. For the UK Biobank study, all people of age between 40 and 69, registered at the National Health Service (NHS) and living within 25 miles from any of the 22 assessment centers, were invited to participate. Almost 9.2 million invitations were sent by mail to recruit the participants (response rate of 5.47%) [12].

Ethical approval for the UK Biobank study was granted by the North-West Multicenter Research Ethics Committee (permission UKB 57519). The Regional Ethics Committee of Uppsala (Sweden) approved the use of UK Biobank data for the present study (2017/198).

### 2.1. Structural MRI Analyses

The gray matter volumes of the brainstem and of the different sub-regions of the cerebellum were obtained via T1 structural MRI using the FAST segmentation tool [13]. Brain images were acquired using 3T Siemens Skyra (software platform VD13), with standard Siemens 32-channel RF receive head coil [14]. All the volumes are expressed in mm${}^{3}$. Complete information regarding MRI data acquisition and processing can be found at the following link: https://biobank.ctsu.ox.ac.uk/crystal/crystal/docs/brain_mri.pdf (accessed on 1 June 2023).

### 2.2. Outcome Variables and Covariates

The main outcome variables investigated in our study are the gray matter volumes of the overall brainstem and cerebellum, as well as all the main sub-regions of the cerebellum.

To decrease the level of bias due to confounding, we conditioned our statistical models within the levels of several important sociodemographic and biological covariates.

All the medical conditions were identified using the variable “Diagnoses—ICD10”. This variable contains information on all the diagnoses that a patient has received, retrieved from all the hospital inpatient records and coded according to the International Classification of Disease 10 (ICD10). For the purpose of the present study, we considered a migraine case as a patient with any diagnosis of migraine (G43).

We also considered a broad set of comorbidities and other disorders that might confound our results. For these reasons, we considered the following health conditions: viral and bacterial infections of the nervous system (A80, A81–A85, A87, A88, G00, G02–G06), diabetes (E10–E14), diseases of the nervous system (G10–G13, G21, G23–G25, G30–G32, G36, G37), mental and behavioral disorders due to psychoactive substances (F10–F19), psychiatric, mental, and behavioral disorders (F00–F02, F05–F07, F20, F22, F23, F25, F30–F34, F38, F40–F45, F48, F50, F53, F54, F62, F63, F68, F99), developmental disorders (F70–F73, F78–F81, F84, F88, F89), epilepsy and sleep disorders (G40, G41, G47, F51), muscle disorders (G56, G71–G73, G80–G83), headaches other than migraine (G44), neuropathies (G50–G55, G57–G63, G70, G90), brain and spine malformations/abnormalities (G91, G93–G97, G99, Q00, Q01, Q03, Q07), cerebrovascular diseases (I60–I63, I65–I69, G45, G46), head and spine injuries and fractures (S001, S007–S010, S01, S02, S020–S024, S026–S029, S04, S06–S09), cardiovascular diseases (I00–I02, I05–I13, I15, I20–I28, I30–I37, I39, I40–I52, I70–I74, I77–I80, I82–I89, I95, I97, I98), and brain cancers (C70–C75, D32, D33, D43).

Other biological and health-related variables that we took into account for the statistical analyses were sex, body mass index (BMI), diastolic blood pressure, age, ethnic background, current tobacco smoking, and alcohol intake frequency. The age is the one registered when the participants visited the assessment center for the MRI scan. The BMI was estimated by impedance measurement. The ethnic background was asked during the initial assessment center interview. The current tobacco smoking status and alcohol intake frequency were also assessed via a touchscreen questionnaire.

We also considered sociodemographic variables such as the assessment center and the indices of multiple deprivation (IMDs). The assessment center variable contains information on which center was visited by each participant. The indices of multiple deprivation is a measurement of poverty in small areas, widely used in the United Kingdom. The IMDs comprise several domains of deprivation, such as income, health, employment, crime, education barriers to housing and services, and living environment.

### 2.3. Statistical Analyses

We used descriptive statistics to summarize the general biological and sociodemographic features of the two study arms. Table 1 displays the results. We also calculated mean, median, standard deviation (SD), and interquartile range (IQR) for cerebellar sub-regions and brainstem volumes. In Table 2, we report the results.

The aim of this study is to estimate the mean volumetric differences between individuals with migraine and controls at the level of the brainstem and cerebellum. Therefore, we used a generalized linear regression model to estimate the volumetric difference parameter. The outcome variable is the volume of a brain region and is expressed in mm${}^{3}$. This can allow for a more straightforward biological evaluation of the results. The main predictor is the variable that encodes the diagnosis information (0 refers to controls and 1 to migraine). We conditioned the model within the levels of other predictors to reduce the bias due to confounding. For the choice of the appropriate set of predictors to include in our model, we used causal directed acyclic graphs (cDAGs). In particular, we included in the cDAG the body mass index (BMI) [15,16,17], age [18,19,20], sex, total brain volume (normalized for head size), diastolic blood pressure, alcohol intake frequency, current tobacco smoking, assessment center, indices of multiple deprivation (IMDs), ethnic background, and all the disorders specified in Section 2.2. The cDAG for our study is portrayed in Figure 1.

In our causal model, migraine is considered the exposure and the brain morphological alteration the outcome. According to this model, the minimal sufficient adjustment required is to condition the regression model within the levels of age, alcohol intake, assessment center, BMI, comorbidities and other disorders, current tobacco smoking, diastolic blood pressure, ethnic background, IMDs, sex, and total volume.
$$Y=\alpha +\beta_{mig}X_{mig}+\beta_{age}X_{age}+\beta_{alc}X_{alc}+\beta_{AssC}X_{AssC}+\beta_{BMI}X_{BMI}+\beta_{com}X_{com}+\beta_{smok}X_{smok}+\;+\beta_{diasBP}X_{diasBP}+\beta_{eth}X_{eth}+\beta_{IMD}X_{IMD}+\beta_{sex}X_{sex}+\beta_{totVol}X_{totVol}+\varepsilon$$

*Y* represents the overall gray matter volume of the cerebellar sub-region or the brainstem. We interpolated this model with the cohort data to obtain an estimation of the regressor $\beta_{mig}$. We also estimated the standard error (SE) and the confidence interval (CI), setting a general confidence level at 95%. We did not correct our confidence level for the multiplicity problem, and therefore our results are to be considered exploratory. We checked the most important model assumptions [21].

We performed statistical inference on the basis of estimation statistics, which is a much more powerful and informative tool than hypothesis testing [22,23,24]. Accordingly, no statistical test was performed and therefore no measure of statistical significance was reported [25,26,27,28]. Furthermore, we calculated Cohen’s d to assess the relative magnitude of the effect sizes that we estimated [29]. We used the formula [30]
$$d=\frac{t(n_{1}+n_{2})}{\sqrt{n_{1}n_{2}}\sqrt{df}}$$
where *t* is the difference between the means ($\beta_{mig}$) divided by the corresponding standard error, $n_{1}$ and $n_{2}$ are the sample sizes of migraineurs and controls, respectively, and df are the degrees of freedom for the t value, i.e., $df=n_{1}+n_{2}-2$. To interpret the values of d thus obtained, we referred to the usual classification, as proposed by Cohen: small (d=0.2), medium (d=0.5), and big effect (d=0.8) [29,30,31].

Moreover, we calculated the relative error (expressed in percentage) to give a quantitative measure of the precision level of our estimations
$$\varepsilon_{r}=\frac{SE}{\beta_{mig}}\times 100\%$$

After the exclusion of the participants without an MRI brain scan, the final sample comprised 712 participants with migraine and 45,681 healthy controls. We ran the generalized linear regression models to estimate the mean difference in gray matter volumes at the cerebellar and brainstem level between the two target populations. Researchers of the UK Biobank study subdivided the cerebellum into the ten different lobular areas, i.e., subregions I–IV, V, VI, crus I and II, VIIb, VIIIa and VIIIb, IX, and X. For each region, they reported the right, left, and vermis volumes (where applicable according to standard anatomy). We summed these values together to study the overall gray matter volumetric differences. In the case of the brainstem, they measured the overall gray matter volume. We thus report the results of our analyses according to this division.

As a secondary analysis, we aimed at estimating the co-variation of the various cerebellar sub-regions. Therefore, we performed a multivariate multiple linear regression, where the outcome variable was a vector Y containing all the cerebellar sub-regions. We included in this model the same predictors of the univariate multiple linear regression. From this model, we extracted the covariance matrix and converted it into the correlation matrix.

All statistical analyses were conducted using R and RStudio (R version 4.1.1 [64 bits], RStudio version 1.4.1106). The complete script used for data curation and the statistical analyses is available on GitHub at the following link: https://github.com/OresteAffatato/Cerebellum_Brainstem_Migraine_Project (accessed on 1 June 2023).

## 3. Results

In Table 2, we report the general descriptive statistics for brainstem and cerebellar volumes. In general, within each group (case or control), we can observe that the mean and median are very similar. Furthermore, the distributions of the volumes of each region are characterized by high dispersion, as we can see from the high values of the standard deviations and the wide IQR. Comparing individuals with migraine and controls, we observe that, also between these groups, the means and medians are very similar. Based on the wide dispersion that is also seen here, we conclude that the distributions of volumes for each brain region of cases and controls have extensive overlapping.

Table 3 displays the results regarding volumetric differences between migraine cases and controls in the investigated cerebellar brain regions and brainstem. Of note, individuals with migraine manifested larger volumes than controls in the sub-regions V (mean difference: 72 mm${}^{3}$, 95% CI [13, 132]), crus I (mean difference: 259 mm${}^{3}$, 95% CI [9, 510]), VIIIa (mean difference: 120 mm${}^{3}$, 95% CI [0.9, 238]), and X (mean difference: 14 mm${}^{3}$, 95% CI [1, 27]).

In most cases, i.e., the brainstem and the majority of the cerebellar sub-regions, estimations are characterized by small standardized effect sizes, as measured by the Cohen’s d, and by low precision, as measured by SE and $\varepsilon_{r}$.

Figure 2 portrays a forest plot of the main results for the cerebellar sub-regions. Even though the estimates are characterized by different levels of precision, we can see that, overall, migraineurs tend to have larger volumes than controls at the level of the cerebellar sub-regions. Individual with migraine appear to have a larger overall cerebellar volume than controls (mean difference: 771 mm${}^{3}$, 95% CI [−87, 1630]). Even though this estimate is less precise than the others at the sub-regional level, it is reasonable to assume that the overall increase in the cerebellar volume seen in individuals with migraine is a result of increases seen at the sub-regional level.

Figure 3 displays the correlation matrix of the multivariate multiple linear regression. All the cerebellar sub-regions appear to be positively correlated with each other. The correlation coefficients vary from a minimum of 0.34 to a maximum of 0.84. The correlation coefficients tend to be larger between adjacent sub-regions and overall confirm the same pattern observed above, i.e., at the cerebellar sub-regional level, individuals with migraine manifest larger volumes than controls.

## 4. Discussion

To the best of our knowledge, this is the first study that has estimated volumetric brain differences between individuals with migraine and controls at the cerebellar and brainstem level in a considerably large cohort applying confounder-adjusted quantitative methods. Notably, we found larger gray matter volumes in the sub-regions V (mean difference: 72 mm${}^{3}$, 95% CI [13, 132]), crus I (mean difference: 259 mm${}^{3}$, 95% CI [9, 510]), VIIIa (mean difference: 120 mm${}^{3}$, 95% CI [0.9, 238]), and X (mean difference: 14 mm${}^{3}$, 95% CI [1, 27]). We were also able to show that the cerebellar sub-regions are characterized by a medium-to-high gradient of positive volumetric correlation, conditioning the model within the levels of a broad range of important biological covariates. The results of this MRI study provide evidence, albeit indirect, of the involvement of the cerebellum in migraine pathophysiology.

Larger gray matter volumes at the cerebellar level might be associated to abnormal brain activity. Other studies on migraine showed decreased functional connectivity between the lobule VIII and brain regions known to be involved in migraine pathogenesis [32]. In particular, studies observed decreased functional connectivity between the primary somatosensory cortex and the ipsilateral sub-region VIIIb and between the right dorsal premotor cortex and the ipsilateral lobule VIII in migraine patients compared to a healthy control [32,33,34]. Therefore, larger volumes in the cerebellar sub-regions (especially in the lobule VIII) might be the consequence of a compensatory mechanism for the altered functional connectivity with sensorimotor regions. This in turn might affect multisensory and pain processing, thus resulting in increased migraine susceptibility [33]. Moreover, abnormal activity at the level of the cerebellum might contribute to an increased activity of calcitonin gene-related peptide (CGRP), a neuropeptide generally found in high concentration in the cerebellum [11]. The CGRP is one of the most important signaling molecules involved in nociception in the trigeminal system, is highly expressed in the cerebellum, and plays a major role in migraine pathogenesis [1,35,36,37]. Abnormal activity and modulation of CGRP in the cerebellum might thus imply increased migraine susceptibility [32].

Previous studies observed that migraine patients have a lower volume at the level of the brainstem than healthy controls [6,7,33,38]. The brainstem is the major center for pain processing and modulation of the trigeminal system and therefore a dysfunction at the level of the brainstem might contribute to migraine onset [33]. We observed volumetric differences that were not significantly large at the level of the brainstem, as all instances of Cohen’s d were close to zero.

It is currently not known to what extent the quantitative shifts in the volume of the cerebellum or brainstem are associated with putative functional dysregulations. We considered and discussed the largest estimated differences between the two target populations, assessed with both the absolute difference and the Cohen’s d, and the most precise, considering the SE and the $\varepsilon_{r}$ as measures of absolute and relative precision, respectively. Overall, our analyses showed that the mean volumetric differences at the cerebellar and brainstem level between the populations of individuals with migraine and controls are generally very small. This can be inferred both from the beta parameters of the regression model, the absolute mean difference, and from the Cohen’s ds, as a measure of the relative difference, normalized by the standard error. In particular, all the Cohen’s ds that we calculated are very close to zero, with the exception of the cerebellar sub-regions V, crus I, VIIIa, VIIIb, and X, even though the present study does not allow us to conclude anything about the real biological meaning and significance of these differences. The presented results constitute only a first step toward a biologically sound evaluation of the volumetric differences and their clinical implications.

A strength of the present study is the application of regression methods to estimate the mean volumetric differences. This is a powerful tool that can allow for an estimation of the population parameter in biologically meaningful units of measure and for more complex and less biased analyses by the inclusion of a wide range of important predictors. Previous studies focused mostly on standardized dimensionless or non-parametric tests, which cannot allow for a clear interpretation of the effect size. We provided estimates expressed in cubic millimeters, which can be easily interpreted. Moreover, we also provided a more thorough assessment of the estimate effect sizes and relative precision via the Cohen’s d and the relative error.

Another strength of our study is the large sample size, which is two orders of magnitude larger than all the previous studies that have addressed the same research question. We were also able to perform a thorough model creation, including several important predictors in our models, through the comprehensive set of variables provided by the UK Biobank. In particular, we used the official ICD10 diagnosis variable, directly retrieved from the inpatient hospital registries. Through a complete and thorough assessment of important comorbidities and brain-related health conditions, we were able highly reduce the level of bias due to confounding.

A major limitation of our study is the cross-sectional setting. For this reason, we were not able to assess any causal relationship. Another limitation of the present study is the average age of the UK Biobank cohort. This cohort comprises mostly elderly people, the majority being older than 60 years, whereas migraine is known to be most prevalent in the younger part of the general population [39]. Therefore, our results might be valid for a population of individuals with a life history of severe migraine (due to the diagnosis at the hospital) rather than younger people with ongoing morbidity. Moreover, the UK Biobank study is characterized by an uncommonly low response rate (almost 6%) [12]. Therefore, our findings can hardly be generalized, as it is difficult to establish a plausible direction of the bias. Another limitation of our study is the high variance of the estimates, which is probably due to some feature of the UK Biobank sampling process, and to the data collection.

## 5. Conclusions

The cerebellum and the brainstem are two major structures of the brain, deeply involved in mechanisms of pain processing. We found larger gray matter volumes in sub-regions V, crus I, VIIIa, VIIIb, and X of the cerebellum in individuals with migraine than controls. Moreover, many anatomically proximal sub-regions manifested high positive volumetric correlation. Future studies have to further elucidate to what extent these volumetric differences in the cerebellum are in order to be considered biologically significant and how they may reflect changes in important cerebellar signaling pathways that are implicated in migraine susceptibility.

## Figures and Tables

**Figure 1 biomedicines-11-02528-f001:**
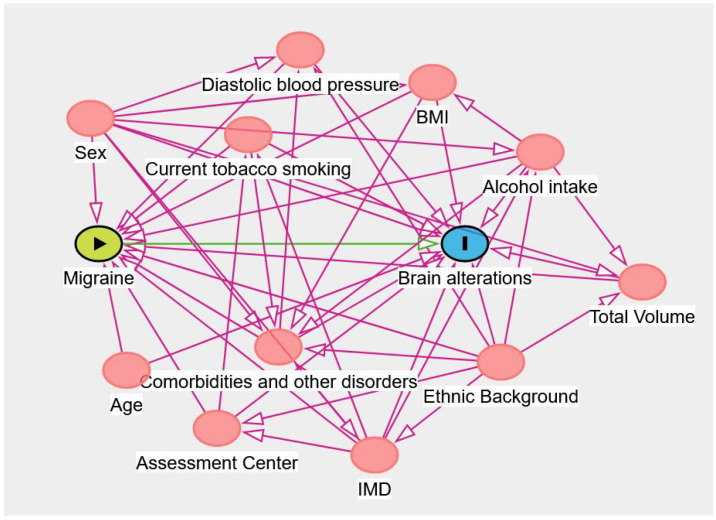
Causal directed acyclic graph (cDAG) representing our causal assumptions for the model. In red are represented all the nodes that produce confounding. The cDAG was drawn using DAGitty v3.0.

**Figure 2 biomedicines-11-02528-f002:**
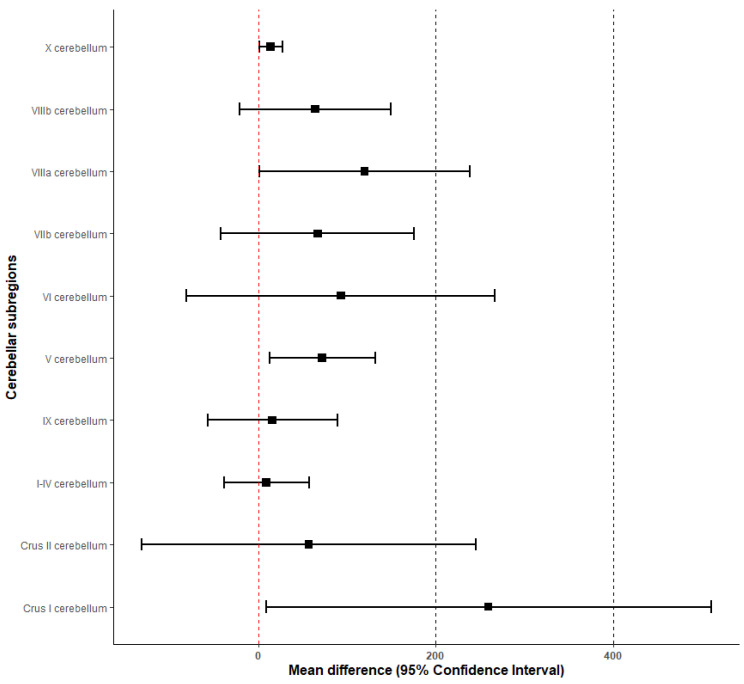
Forest plot displaying the mean differences and the corresponding 95% CIs for each cerebellar sub-region.

**Figure 3 biomedicines-11-02528-f003:**
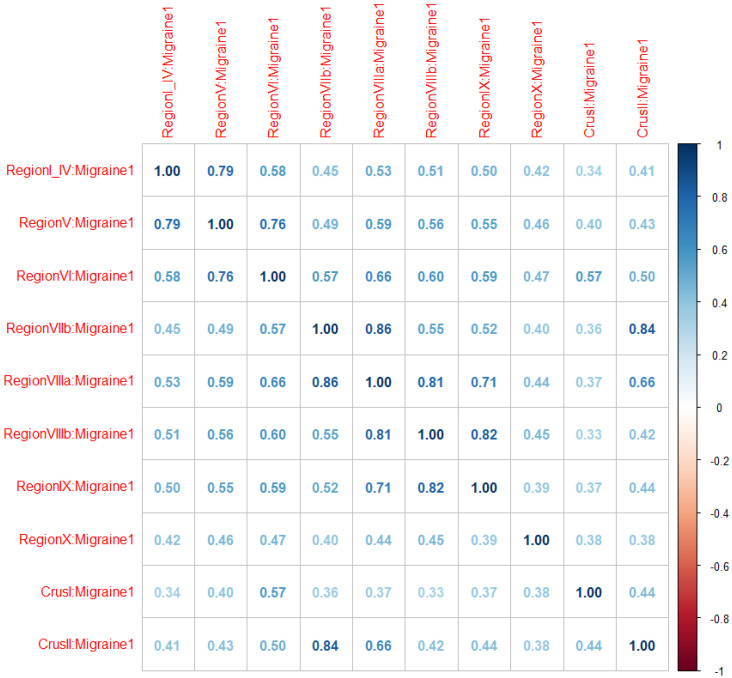
Correlation matrix extracted from the multivariate multiple linear regression. Here, the correlation coefficients between the difference cerebellar sub-regions are reported, with regard to migraine diagnosis, adjusting for all the other covariates.

**Table 1 biomedicines-11-02528-t001:** Descriptive statistics of main biological and sociodemographic factors.

	Migraineurs	Controls
	N = 712	N = 45,681
**Sex**		
Women	527 (74%)	23,977 (53%)
Men	185 (26%)	21,704 (47%)
Age (mean ± SD ^1^)	63 ± 8	64 ± 8
BMI (mean ± SD ^1^)	27 ± 5	27 ± 4
IMD (median, IQR ^2^)	12.4, [7.1, 20.5]	11.3, [6.7, 19.9]
**Current tobacco smoking**		
Yes, on most or all days	10 (2%)	616 (2%)
Only occasionally	14 (2%)	911 (2%)
No	682 (96%)	43,825 (96%)
Prefer not to answer	0 (0%)	8 (0%)
**Alcohol intake frequency**		
Daily or almost daily	60 (8%)	7678 (17%)
Three or four times a week	136 (19%)	12,786 (28%)
Once or twice a week	174 (25%)	11,976 (26%)
One to three times a month	97 (14%)	5236 (12%)
Special occasions only	135 (19%)	4725 (10%)
Never	104 (15%)	2942 (7%)
Prefer not to answer	0 (%)	17 (0%)

^1^ Standard deviation. ^2^ Interquartile range.

**Table 2 biomedicines-11-02528-t002:** Descriptive statistics for the gray matter volumes of brainstem and all cerebellar lobular sub-regions displayed for cases and controls. We calculated mean, median, standard deviation (SD), and interquartile range (IQR). All measures are expressed in mm${}^{3}$.

Brain Region	Migraineurs (N = 712)	Controls (N = 45,681)
Cerebellum	Mean = 89,511, Median = 89,482	Mean = 90,346, Median = 90,472
	SD = 10,564	SD = 11,158
	IQR = [82,646, 96,887]	IQR = [83,227, 97,605]
I–IV cerebellum	Mean = 3899, Median = 3881	Mean = 3965, Median = 3940
	SD = 550	SD = 575
	IQR = [3518, 4228]	IQR = [3567, 4328]
V cerebellum	Mean = 5174, Median = 5134	Mean = 5230, Median = 5199
	SD = 707	SD = 732
	IQR = [4710, 5623]	IQR = [4728, 5697]
VI cerebellum	Mean = 14,688, Median = 14,761	Mean = 14,855, Median = 14,841
	SD = 2126	SD = 2201
	IQR = [13,251, 16,206]	IQR = [13,385, 16,299]
Crus I cerebellum	Mean = 21,819, Median = 21,766	Mean = 21,952, Median = 21,838
	SD = 3145	SD = 3230
	IQR = [19,688, 23,883]	IQR = [19,764, 24,026]
Crus II cerebellum	Mean = 16,012, Median = 15,815	Mean = 16,190, Median = 16,133
	SD = 2196	SD = 2307
	IQR = [14,631, 17,409]	IQR = [14,673, 17,648]
VIIb cerebellum	Mean = 8037, Median = 8041	Mean = 8114, Median = 8109
	SD = 1217	SD = 1322
	IQR = [7222, 8810]	IQR = [7280, 8960]
VIIIa cerebellum	Mean = 8521, Median = 8507	Mean = 8569, Median = 8624
	SD = 1382	SD = 1473
	IQR = [7641, 9428]	IQR = [7649, 9540]
VIIIb cerebellum	Mean = 5761, Median = 5730	Mean = 5837, Median = 5818
	SD = 1018	SD = 1049
	IQR = [5017, 6443]	IQR = [5119, 6515]
IX cerebellum	Mean = 4433, Median = 4400	Mean = 4463, Median = 4409
	SD = 830	SD = 866
	IQR = [3821, 5007]	IQR = [3850, 5015]
X cerebellum	Mean = 1167, Median = 1170	Mean = 1172, Median = 1168
	SD = 165	SD = 166
	IQR = [1059, 1274]	IQR = [1062, 1278]
Brainstem	Mean = 4834, Median = 4755	Mean = 4857, Median = 4794
	SD = 797	SD = 848
	IQR = [4293, 5323]	IQR = [4301, 5333]
Brain volume	Mean = 1,503,249, Median = 1,500,440	Mean = 1,490,284, Median = 1,490,310
	SD = 74,378	SD = 73,870
	IQR = [1,451,903, 1,556,158]	IQR = [1,439,020, 1,541,320]

**Table 3 biomedicines-11-02528-t003:** Mean gray matter volumetric differences between individuals with migraine and controls. Volumes all measured in mm${}^{3}$.

Brain Region	Mean ± SE	95% Confidence Interval	$\varepsilon_{r}$	Cohen’s d
Cerebellum	771 ± 438	[−87, 1630]	57%	0.08
I–IV cerebellum	9 ± 24	[−38, 57]	267%	0.02
V cerebellum	72 ± 30	[13, 132]	42%	0.10
VI cerebellum	93 ± 88	[−81, 266]	95%	0.05
Crus I cerebellum	259 ± 128	[9, 510]	49%	0.08
Crus II cerebellum	57 ± 96	[−131, 245]	168%	0.03
VIIb cerebellum	67 ± 55	[−42, 175]	82%	0.05
VIIIa cerebellum	120 ± 60	[0.9, 238]	50%	0.09
VIIIb cerebellum	64 ± 44	[−21, 149]	69%	0.06
IX cerebellum	16 ± 37	[−57, 89]	231%	0.02
X cerebellum	14 ± 7	[1, 27]	50%	0.09
Brainstem	27 ± 32	[−36, 89]	119%	0.04

## Data Availability

The data that support the findings of this project are available from UK Biobank. All data generated during this study are included in this paper.

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
