# Peer review of "Volumetric Differences in Cerebellum and Brainstem in Patients with Migraine: A UK Biobank Study"

_biomedicines, 2023, doi:10.3390/biomedicines11092528_

Round 1
Reviewer 1 Report (Previous Reviewer 1)
The study has several crucial issues that limit its value.
All the medical conditions have been identified using the variable "Diagnoses - ICD10". This variable contains information on all the diagnoses a patient had received, retrieved from all the hospital inpatient records and coded according to the International Classification of Disease 10 (ICD10).
Only cases with purposefully performed MRI scans were included.
Mostly older (above 60) adults compose the UK Biobank cohort.
Considering the above, the participant set of migraineurs is nor representative of the general population with migraine. Older adults that had to undergo MRI scans and hospitalization for headache do not constitute a representative sample of migraineurs (the diagnosis is uncertain, in case of correct diagnosis this subgroup consists of very serious cases with migraine and probably neurological semiology).
Misclassification is certain (reflected on the prevalence of migraine among responders), reverse causality is possible, multiple comparisons were not adjusted.
Author Response
We would like to thank the reviewer for the precise concerns that has been raised. The reviewer highlights several limitations that we are aware of and most of them are intrinsic to the cohort. This presents some limitations in generalizability or external validity, however it does not hinder the internal validity of our study. Here below the reviewer can find the answers point by point.
- Considering the above, the participant set of migraineurs is nor representative of the general population with migraine. Older adults that had to undergo MRI scans and hospitalization for headache do not constitute a representative sample of migraineurs (the diagnosis is uncertain, in case of correct diagnosis this subgroup consists of very serious cases with migraine and probably neurological semiology).We do agree, this sample is not representative of the target population, which is a limitation that we addressed in our limitation section (lines 295 - 303). As we discussed in that section, we cannot generalize our results, even though the results could be generalized to a target population that more closely resembles our sample, i.e. people with life history of severe migraine. Moreover, this is hardly an issue when it comes to the internal validity of the study, as it has been pointed out by many epidemiologists. In particular, we refer to Modern Epidemiology 4th Edition by Lash et al., and to a couple of papers in the International Journal of Epidemiology: https://academic.oup.com/ije/article/42/4/1012/656034?login=false and https://academic.oup.com/ije/article/42/4/1014/658592?login=false . We provided thorough statistical analyses that controlled for the most important confounding factors and used one of the largest cohort-based studies available at the moment. Therefore, we are confident that our results are sound, given of course all the limitations.
- The diagnosis is uncertain, in case of correct diagnosis this subgroup consists of very serious cases with migraine and probably neurological semiologyWe agree with the reviewer also in these regards. UK Biobank offers two major and straightforward ways of assessing the diagnosis: the ICD10 official diagnosis or the self-reported diagnosis. In a previous version of our study, we adopted the self-reported diagnosis, therefore including much more cases than the present version. However, this choice was criticized by a previous reviewer as it might introduce much more bias to have the diagnosis just reported by the individual and not officially reported by a medical doctor. Therefore, we decided to adopt the other way, the official ICD10 diagnosis from the hospital registries. In this way we are much more sure about the correct diagnosis, however, as the reviewer pointed out, these are most likely much more severe cases of migraine. This is a methodological choice, and we believe that it is more rigorous at this point to keep the diagnosis from the medical expert and then make an inference only on the most severe cases rather then using the self-reported diagnosis and introduce more bias. To be clearer, according to your suggestion, we modified the limitation section in these regards, lines 298-299.
- Misclassification is certain (reflected on the prevalence of migraine among responders). This bias is present, however we don't believe this this might play a major role. Indeed people with non-severe migraine might have been considered controls instead of cases. However, we adjusted the model within the levels of several neurologic comorbidities, including other headache syndromes, and other migraine comorbidities; therefore we should have partially accounted for this problem.
- Reverse causality is possible. We do agree with this point and we highlighted this in the limitation section, lines 293-294.
- Multiple comparisons were not adjusted. We are aware of this issue and we discussed it in the Method section, lines 162-164. It has to be said that the multiplicity problem is not necessarily a major issue, as pointed out by Gelman and others in the book Regression and other stories. One could also check: https://www.tandfonline.com/doi/abs/10.1080/19345747.2011.618213

Reviewer 2 Report (Previous Reviewer 2)
I am sattisfied! The paper is good!
Author Response
We kindly thank you for your appreciation!
Reviewer 3 Report (New Reviewer)
Thank you for your important paper on the volumetric anlysis of brain stem and celleberum among migraine or non-migraine individuals. They concluded cerebellum gray matter is fewer in migraine patients than healthy control
Abstract Good. If you can, please provide the year of the dataset.
Introduction or discussion, please mention dentatothalamo cortical patheway which is related to take an action as initial. https://pubmed.ncbi.nlm.nih.gov/33767888/
Why did you use a generalized linear model instead of a generalized linear mixed model? Why not use the ID of the data or the hospital of the MRI acquired as random variables?
There are several well-known mechanisms, such as the hypothalamic generator hypothesis, the descending inhibitory system of the PAG, and cortical depression in the occipital lobe, but the cerebellum is new to me. Since we are at this point, why don't you further discuss the pathophysiology of migraine and summarize what theories are out there?
In particular, is it related to chronic migraine or medication-overuse headache?
Author Response
We thank the reviewer for the insightful comments. Here below our answers:
- Abstract Good. If you can, please provide the year of the dataset. We think that the year of the dataset is not well-defined. The UK Biobank project started in 2006 and it comprises several waves, and we used data from several of those.
- Introduction or discussion, please mention dentatothalamo cortical patheway which is related to take an action as initial. https://pubmed.ncbi.nlm.nih.gov/33767888/. We had a look at the suggested reference. It is a case report on a 92 years old woman who experienced bilateral thalamic infarction, which then developed several symptoms, including persistent mutism. We do not think that is specific condition nor the evidence level of the paper is appropriate or related to our topic. We are anyway grateful for the suggestion.
- Why did you use a generalized linear model instead of a generalized linear mixed model? Why not use the ID of the data or the hospital of the MRI acquired as random variables? We thank the reviewer for the interesting suggestion. We used the generalized linear regression as it is a standard tool to estimate the mean difference between two target populations, as in our case. To incorporate random the variables you mentioned as random effects wouldn't make much of a difference according to us. Also we believe that a variable as the Assessment Center where the MRI scan was taken would have a deterministic effect on our outcome of interest, because the fact of living close to the Assessment Center depends strongly on the socioeconomic status, which we know has a major influence on migraine diagnosis and on the health in general.
- There are several well-known mechanisms, such as the hypothalamic generator hypothesis, the descending inhibitory system of the PAG, and cortical depression in the occipital lobe, but the cerebellum is new to me. Since we are at this point, why don't you further discuss the pathophysiology of migraine and summarize what theories are out there? We are aware of the other theories and their evidences, however discussing all of them would fall rather out of the scope of the present work. Our results are discussed in the light of the action of the CGRP which we believe is the most plausible and inherent explanation when it comes to the brainstem. We mention further explanatory evidence in the lines 239 - 254.
- In particular, is it related to chronic migraine or medication-overuse headache? For the purpose of the present study we considered a case of migraine a person having any diagnose of migraine, as specified by the ICD 10. The exact details are in lines 103 - 107.
Reviewer 4 Report (New Reviewer)
The aim of this study was to investigate possible associations between the morphology of cerebellum andbrainstem and migraine, focusing on gray matter differences in these brain areas, bearing in mind the contradictory data reported in the literature on this topic. The work is conceptualized in a coherent and detailed way, and addresses possible methodological and conceptual biases with a critical approach. The organizational effort to overcome sampling bias was truly remarkable. The choice of adopting a generalized linear regression model seems appropriate. To reduce generic confounding bias Authors conditioned their statistical models within the levels of several important sociodemographic and biological covariates (lines 101-102). Lines 163: Authors report that results are to be considered exploratory , and this opinion is repeated and underlined several times In Figure 1 Casual directed acyclic graphs is well done and self explaining The discussion is articulated and well conducted, and attempts to respond to all the inconsistencies that emerge with the literature data. Globally the data show migraineurs tend to have larger volumes than controls, and the sum of the differences, even if very small, proceeds the final aspect. In figure 3 the correlation matrix shows some significant differences in delimited areas. Lines 265-8: Overall, our analyzes showed that the mean volumetric differences at cerebellar and brainstem level between the populations of individuals with migraine and controls are generally very small. This sentence is really important. Lines 272-5: I agree with this conclusion that should be moved in the conclusion section. Lines 301-302: move this sentence to the conclusion section.
Author Response
We are grateful to the reviewer for the nice comments. We modified the conclusion section to comply to the recommendation: Lines 272-5: I agree with this conclusion that should be moved in the conclusion section. Lines 301-302: move this sentence to the conclusion section.
Round 2
Reviewer 1 Report (Previous Reviewer 1)
Unfortunately, there are issues that cannot be corrected due to the original design of the study. No corrections were made - a second review round is not necessary
Author Response
As we said, we agree with the reviewers and we are well aware of the raised concerns. We appreciate the comments.
This manuscript is a resubmission of an earlier submission. The following is a list of the peer review reports and author responses from that submission.
Round 1
Reviewer 1 Report
This is a very interesting article on grey matter differences in cerebellum and brainstem of migraineurs and healthy controls. It is a based on a large population-based sample and implemented a strict eligibility strategy that ensured the exclusion of subjects with comorbidities that could impact the morphology of the brain. Please make sure that the term ‘‘migraineur’’ is replaced with periphrastic terms (e.g., individuals with migraine) throughout the text.
The main drawbacks of the current research are the following:
The diagnosis of the health conditions of the participants was self-reported. Although such an approach is usually acceptable in the case of covariates, the main exposure and outcome have to be more subjectively assessed. The outcome of the current analysis fulfills this condition. However, the identification of migraine cases is problematic: not only overdiagnosis should be expected among ‘‘hypochondriac’’ individuals but also under-reposting among ‘‘true sufferers’’ (which is the reflected in the low prevalence of migraine in the analysed sample). Secondly, although the authors present the aforementioned strategy as advantageous, given the potential inclusion of currently affected (at the time of visiting the Assessment Centre) individuals, in fact it constitutes a major drawback considering the misclassification of the participants with long-standing migraine during middle age (i.e., the majority of individuals with migraine worldwide). These drawbacks should be clear for the readers to assess for themselves.
More information is required about the UK Biobank study. Selection process? Random? Age ranges? Sex and age stratification? Etc… Although this information may be readily available in other articles, each study should stand by itself.
‘‘In general, within each group (case or control), we can observe that mean and median are very similar, therefore we can infer that the volumes are normally distributed.’’ Normal distribution should be statistically (or graphically) confirmed using conventional approaches. Mean -Median similarities in the absence of bell curve distributions etc are not enough. Please provide a short description of the assumptions that you have checked (for normality and main analysis).
Have you considered a multivariate general linear model? Your model does not account for inter-correlations among different cerebellum and brainstem regions. Two multivariate analyses would capture these inter-correlations and limit the possibility of trivial associations.
In any case, according to your approach only VIIIb cerebellum regions differ (not VIIIa).
Author Response
This is a very interesting article on grey matter differences in cerebellum and brainstem of migraineurs and healthy controls. It is a based on a large population-based sample and implemented a strict eligibility strategy that ensured the exclusion of subjects with comorbidities that could impact the morphology of the brain. Please make sure that the term ‘‘migraineur’’ is replaced with periphrastic terms (e.g., individuals with migraine) throughout the text.
We would like to kindly thank the reviewer for all the insighful comments. With regards to the «migraineur» terminology, we replaced it as suggested throughout the text.
The main drawbacks of the current research are the following:
- The diagnosis of the health conditions of the participants was self-reported. Although such an approach is usually acceptable in the case of covariates, the main exposure and outcome have to be more subjectively assessed. The outcome of the current analysis fulfills this condition. However, the identification of migraine cases is problematic: not only overdiagnosis should be expected among ‘‘hypochondriac’’ individuals but also under-reposting among ‘‘true sufferers’’ (which is the reflected in the low prevalence of migraine in the analysed sample). Secondly, although the authors present the aforementioned strategy as advantageous, given the potential inclusion of currently affected (at the time of visiting the Assessment Centre) individuals, in fact it constitutes a major drawback considering the misclassification of the participants with long-standing migraine during middle age (i.e., the majority of individuals with migraine worldwide). These drawbacks should be clear for the readers to assess for themselves.
We agree with the reviewer that it is important that exposure and outcome should be precisely measured. This is usually the case for smaller cohorts, designed with a well-defined target diagnosis. In the case of UK Biobank, this is a general limitation, as it is a very large cohort, and not all the variables are measured with the same degree of precision. The purpose of this cohort is to allow the broadest variety of research questions to be tackled using the largest sample size. Nevertheless, the UK Biobank cohort is still widely used to address different health-realted questions, using also the self-reported diagnosis (purely as examples, the reviewer might consider to references DOI: 10.5037/jomr.2014.5302 and DOI: 10.1038/s41598-017-05507-6 ). We do agree that the risk of misclassification is a major drawback of the study, even though we also believe that individuals having migraine are to a certain extent self-aware of their condition, given the severity of this disorder. We believe that the greatest implication of this misclassification would be that our conclusion would apply strictly to the participants that have severe headaches (since we cannot differentiate between, e.g., migraine and tension-type headache) given also a more diluited effect estimation due to the present of misdiagnosed participants. We added a comment in the limitation section of the discussion, in agreement with your suggestion (lines 267 to 273).
- More information is required about the UK Biobank study. Selection process? Random? Age ranges? Sex and age stratification? Etc… Although this information may be readily available in other articles, each study should stand by itself.
Thank you for your comment in this regard. According to your suggestion, we improved a paragraph of the Materials and Methods section (lines 81 to 87).
- ‘‘In general, within each group (case or control), we can observe that mean and median are very similar, therefore we can infer that the volumes are normally distributed.’’ Normal distribution should be statistically (or graphically) confirmed using conventional approaches. Mean -Median similarities in the absence of bell curve distributions etc are not enough. Please provide a short description of the assumptions that you have checked (for normality and main analysis).
We appreciate the technical comment from the reviewer. As a general remark, the problem with the tests for normality would compare the distribution of our data with the hypothesis of (perfect) normality, which is hardly met in any study. Moreover, they would result in the calculation of a p-value which would be used to make a dichotomous statement: significant or not. The problem of this approach is that it introduces a dichotomization that we believe hardly exists in nature. The convergence of a data distribution to the normal curve is a matter of degree, rather than a two-states property (it is either present or not). On the other hand, the graphical check would not differ from our approach. Using Table 2, one could easily see that the difference between mean and median is several orders of magnitude smaller than the two statistics themselves. Therefore, they are quite similar. This implies the symmetry of the distribution. Morevover, the Central Limit Theorem guarantees that the volume of the region tends towards the normal (as each target volume is the sum of two or three random variables, right, left parts and the vermis). We added a brief comment on the assumption check on the methods part (lines 189 onwards).
- Have you considered a multivariate general linear model? Your model does not account for inter-correlations among different cerebellum and brainstem regions. Two multivariate analyses would capture these inter-correlations and limit the possibility of trivial associations. In any case, according to your approach only VIIIb cerebellum regions differ (not VIIIa)
We agree that the multivariate approach would be a possible way to account for inter-correlations between the regions. The approach we choose assumes the independence of the effects, to a certain extent, in the sense that we look at the volumetric difference of each region independently of the other. Risk for trivial association is anyway controlled via the conservative Bonferroni correction applied on confidence intervals, which reduces the type I error rate.

Reviewer 2 Report
An interesting study. For me, your choice of statistics seems excellent!
1. As a reader, I would appreciate a better background description of why there should be a connection between migraine and low gray matter volumes on MRI in two specific areas of cerebellum.
2. Moreover, I wonder, are your findings an expression of a co-incidence or just a co-existence.
3. Row 177-85: Perhaps better placed under ‘Methods’ ?
4. In these days of ‘Woke’, it might be better to use ‘gender’ instead of ‘sex’ ?
5. How do you define brainstem? Mesencephalon and pons? In function, a huge area if you also include the medulla oblongata. How do you explain that you do not have the same findings in the brainstem as in the cerebellum like other studies have shown. Please, give your comments.
6. From row 268 and forward you mention weaknesses of the study, as high age, and low response rate. Especially, I am worried about ‘self-diagnosing’ migraine. According to my clinical experience one must be very careful accepting this. A lot of these patient suffer from tension-type headache. Thus, I wonder could your findings be secondary to many years with unspecified headache and not primarily a cause of migraine? In fact, I am afraid, your material do not allow you to conclude anything about migraine only about headache. At least, you have to be very humble in your conclusion!
Author Response
An interesting study. For me, your choice of statistics seems excellent!
- As a reader, I would appreciate a better background description of why there should be a connection between migraine and low gray matter volumes on MRI in two specific areas of cerebellum.
The second paragraph of the introduction has been expanded to clarify further this aspect. More specific explanations of the underlying mechanisms are left in the discussion part (lines 220 to 235), where we discuss them in the context of our findings.
- Moreover, I wonder, are your findings an expression of a co-incidence or just a co-existence.
To further improve the robustness of our findings we adopted the Bonferroni correction for the confidence intervals, thus decreasing the impact of chance in our study. Moreover, the large sample size decreases the impact of pure chance on our results.
- Row 177-85: Perhaps better placed under ‘Methods’?
As suggested, we moved this paragraph in the “Methods” section. Now, it can be found now at lines 173 to 181.
- In these days of ‘Woke’, it might be better to use ‘gender’ instead of ‘sex’?
The “sex” variable refers to the biological sex, an information taken directly from the central registries (in some cases updated by the participants) of the NHS. We used this as a proxy for all the sex-related features (hormones, genetics…) that in our opinion would introduce confounding. To our understanding, “gender” refers to another feature, on the socio-cultural level, and this information is neither available in the UK Biobank database nor we could find any evidence that this plays any role in the causal or statistical association between migraine and MR brain volumetric differences. Anyway we refer to the definition given by the WHO, here the reviewer can find some more information: https://www.who.int/europe/health-topics/gender#tab=tab_1
- How do you define brainstem? Mesencephalon and pons? In function, a huge area if you also include the medulla oblongata. How do you explain that you do not have the same findings in the brainstem as in the cerebellum like other studies have shown. Please, give your comments.
The UK Biobanks researchers adopted standard atlases for the MRI brain scans, such as the Harvard-Oxford atlas. Our analyses were based on these standardized definitions of brain regions, in the case of brainstem including mesencephalon, pons and medulla oblongata. Indeed, an extensive area in functional terms. On this regards, we just wanted to stress the fact that the MRI scans (as well as the processing analyses) were designed and produced by the UK Biobank researchers. Therefore, we worked on the materials produced by them, with limited possibilities of sub-dividing certain areas. According to our estimates, the brainstem gray matter volume is also lower in migraineurs than controls, thus manifesting an association going in the same direction of the cerebellum. However, the standard error is of the same magnitude of the effect and therefore we decide not to comment on that as in this case the measure is characterized by very low precision. Making a statement in this regards, even though purely speculative, might be misleading for our readers.
- From row 268 and forward you mention weaknesses of the study, as high age, and low response rate. Especially, I am worried about ‘self-diagnosing’ migraine. According to my clinical experience one must be very careful accepting this. A lot of these patient suffer from tension-type headache. Thus, I wonder could your findings be secondary to many years with unspecified headache and not primarily a cause of migraine? In fact, I am afraid, your material do not allow you to conclude anything about migraine only about headache. At least, you have to be very humble in your conclusion!
We totally agree with you in this regards. These concerns were already partially highlighted in the “Selection of the participants” sub-section, lines 90 to 94. However, we do still believe that patients are generally self-aware of their condition, since migraine is such a burdensome and severe disorder, even though, as the reviewer pointed out, we cannot exclude the fact that in some cases the patient has tension-type headache. We added this limitation in the discussion (lines 267 to 273), in agreement with your important comment.

Round 2
Reviewer 1 Report
Thank you for considering my suggestions.